# On the Optimization of Fermentation Conditions for Enhanced Bioethanol Yields from Starchy Biowaste via Yeast Co-Cultures

Mohamed Hashem [1,2,*], Saad A. Alamri [1], Tahani A. Y. Asseri [1], Yasser S. Mostafa [1], Gerasimos Lyberatos [3,4] and Ioanna Ntaikou [3,4]

1   Department of Biology, College of Science, King Khalid University, Abha 61413, Saudi Arabia; saralomari@kku.edu.sa (S.A.A.); tasseri@kku.edu.sa (T.A.Y.A.); ysolhasa1969@hotmail.com (Y.S.M.)
2   Botany and Microbiology Department, Faculty of Science, Assiut University, Assiut 71516, Egypt
3   Zografou Campus, School of Chemical Engineering, National Technical University of Athens, GR 15780 Athens, Greece; lyberatos@chemeng.ntua.gr (G.L.); ntaikou@iceht.forth.gr (I.N.)
4   Institute of Chemical Engineering Sciences, Foundation for Research and Technology, GR 26504 Patra, Greece
*   Correspondence: mhashem@kku.edu.sa; Tel.: +966-172-417-625

**Abstract:** The present study aims to assess the impact of the type of yeast consortium used during bioethanol production from starchy biowastes and to determine the optimal fermentation conditions for enhanced bioethanol production. Three different yeast strains, *Saccharomyces cerevisiae*, *Pichia barkeri*, and *Candida intermedia* were used in mono- and co-cultures with pretreated waste-rice as substrate. The optimization of fermentation conditions i.e., fermentation time, temperature, pH, and inoculum size, was investigated in small-scale batch cultures and subsequently, the optimal conditions were applied for scaling-up and validation of the process in a 7-L fermenter. It was shown that co-culturing of yeasts either in couples or triples significantly enhanced the fermentation efficiency of the process, with ethanol yield reaching $167.80 \pm 0.49$ g/kg of biowaste during experiments in the fermenter.

**Keywords:** bioethanol; starchy wastes; hydrolysate; co-cultures; scale-up

## 1. Introduction

Bioethanol is one of the most important renewable fuels that could alleviate the negative environmental effects generated by the utilization of oil fuels [1] and represents a sustainable energy carrier in the future. Now, there is a great interest in the production of bioethanol as renewable bioenergy by lowering its production cost through the exploitation of low-cost substrates, efficient fermentative organisms, and optimization of the process conditions for obtaining the maximum yield [2]. The most important factor in bioethanol production is the availability and renewability of the biomass. Agricultural and industrial biowastes emerged as an ultimate cost-effective raw material for biofuel production [3,4]. Food waste (FW) is organic waste discharged continuously from restaurants and food processing [5]. Solid and liquid FW are produced daily by the industries of food processing, restaurants, and hotels. These kinds of wastes include foods that are not up to the specified quality control standards, peelings and remnants from crops, fruits, and vegetables and could play a key role in the sustainability of bioethanol production in the near future [6].

To be ready for fermentation, these biowastes should be subjected to hydrolysis of the complex carbohydrates towards simple fermentable sugars. The selection of the most efficient methodology to achieve this could vary significantly for food wastes of different compositions. Generally, more severe pretreatment is required for FW with high ligno-cellulosic content, whereas direct hydrolysis can be applied to starchy FW, leading to efficient saccharification. The saccharification of starchy wastes can be achieved either enzymatically [7–9] or chemically, mainly via dilute acids such as sulfuric [10,11], phosphoric [12], and hydrochloric [13] facilitated by simultaneous thermal treatment. In general,

low temperatures lead to solubilization of starch whereas for its further saccharification temperatures above 90 °C are required. Although acid hydrolysis of starchy substrates constitutes a cost-effective and fast method for their saccharification, there are certain drawbacks that must be taken into account during the application of such a process. These include the need for equipment capable of withstanding corrosion and the possible formation of byproducts that are generated during acidic degradation of sugars, such as furfurals, that might have an inhibitory effect on fermentative microorganisms [11]. To avoid the latter the selection of proper conditions and tolerant microorganisms should be made. In this context the co-culturing of different microbial strains could be proven to be beneficial, resulting in higher tolerance to inhibitors [14] and more efficient utilization and bioconversion of carbon sources due to the potential synergistic action of the metabolic pathways of the involved strains [15].

Indeed, the application of yeast co-cultures for ethanol production has been proposed by many researchers. In general, yeast co-culturing is mainly used when targeting the exploitation of feedstocks that contain both C5 and C6 sugars or polymers of them, such as lignocellulosic biomass. Conventional fermentative yeast *Saccharomyces cerevisiae* cannot utilize pentoses and other sugars other than glucose. So, the released sugars cannot be completely exhausted during the fermentation process, and low productivity of ethanol results. To overcome this challenge, different yeast species capable of fermenting pentoses such as *Pichia stipitis, Candida shehatae, Pachysolen tannophilus,* etc. were selected to be used in co-cultures with *S. cerevisiae* to maximize the bioconversion efficiency of such waste materials [16–19]. Moreover, the co-culturing of *S. cerevisiae* with other yeast strains has been proven to enhance ethanol yields even from other types of sugars, such as sucrose, maltose, and oligosaccharides [20,21].

The goal of the current study was to investigate the production of bioethanol from a starchy biowaste using consortia of yeast strains that were isolated from naturally fermenting waste-rice i.e., they were indigenous of a material similar to the waste that was used as feedstock in the current study. Based on their fermentation efficiency on glucose, three yeast strains were selected, *Saccharomyces cerevisiae* KX008611, *Pichia barkeri* KX008612, and *Candida intermedia* KX008614, and their ethalogenic potency during mono-culturing and co-culturing was investigated using the starchy biowaste, after chemical hydrolysis, as the sole carbon source. The strategy of selecting fermentative microorganisms that are indigenous on substrates similar to the targeted feedstock is considered to be advantageous for the efficiency of the processes to be developed since, as reported by previous studies, they have better performance [22] and tend to be more robust at industrial applications [23] and as such those strains are expected to adapt easily to the new substrate. The effect of key fermentation parameters on ethanol yields was studied and was statistically evaluated and the scaling-up of the process was subsequently evaluated for the monocultures and the different combinations of co-cultures at the optimal conditions. In this context, the novelty and originality of the study lies in (a) proving that the endogenous yeasts of the waste can be quite efficient for its fermentation towards ethanol under optimized conditions, (b) the investigation of the effect of co-culturing versus mono-culturing on the ethanol yields and the fermentation efficiency of the waste, and (c) the confirmation of the results of the proposed methodology during the scaling-up of the process.

## 2. Materials and Methods

### 2.1. Biowaste

The starchy biowaste (SBW) that was used as feedstock in the present study was meal leftovers containing mainly waste-rice (Style Al walimah) and was collected from restaurants as previously described by Hashem et al. [13]. The SBW was ground in a laboratory blender to obtain homogeneous slurry containing particles ≤3 mm and its physicochemical characteristics were assessed as follows: total solids, 0.73 g/g, total starch, 0.54 g/g, reducing sugars, 0.11 g/g, total nitrogen, 58.0 mg/g, and ash content, 49.7 mg/g.

### 2.2. Hydrolysis

The SBW slurry was subjected to thermo-acid hydrolysis as previously reported by Hashem et al. [12]. Briefly, 20% SBW (*w/v*) was acidified with 1 M HCl to adjust the pH to 1. The acidified slurry was then treated at 121 °C for 30 min. After cooling, the pH was adjusted to 5 using 1 M NaOH and the complete hydrolysis of starch in the sample was verified using an iodine color test. The concentration of the reducing sugar after this step was estimated as 0.36 g/g of the pretreated SBW.

### 2.3. Fermentation Experiments

#### 2.3.1. Microorganisms

Fermentation tests were performed using three different yeast strains, *S. cerevisiae* KX008611, *P. barkeri* KX008612, and *C. intermedia* KX008614 that were previously isolated from naturally fermented waste-rice [13]. Yeast strains were identified by sequencing the D1/D2 domain of 26S rDNA regions using the primers NL1 (5′-GCATATCAATAAGCGGA GGAAAAG-3′), and NL4 (5′-GGTCCGTGTTT CAAGACGG-3′) [13] and were submitted to the GenBank as new yeast strains.

The inoculum of each yeast was prepared by growing it in yeast-malt broth medium (YMB) in a rotary shaker (150 rpm) for 48 h at 25 °C.

#### 2.3.2. Optimization of Bioethanol Production in Small-Scale Flask Experiments

The hydrolyzed SBW was used throughout the study as the fermentation medium. Batch experiments were carried out with monocultures and co-cultures of *S. cerevisiae* KX008611, *P. barkeri* KX008612, and *C. intermedia* KX008614. Seven sets of batch experiments were carried out in 250-mL screw-capped bottles with 100 mL of hydrolyzed SBW with an initial sugar concentration of 32 g/L, as follows: 1. *S. cerevisiae* KX008611, 2. *P. barkeri* KX008612, 3. *C. intermedia* KX008614, 4. *S. cerevisiae* KX008611+ *P. barkeri* KX0086125, 5. *S. cerevisiae* KX008611+ *C. intermedia* KX008614, 6. *P. barkeri* KX008612+ *C. intermedia* KX008614, and 7. *S. cerevisiae* KX008611+ *P. barkeri* KX008612+ *C. intermedia* KX008614. Each culture was inoculated with 5% (*w/w*) of pre-culture with a biomass concentration of ~$10^6$ cell/mL. Three replicate cultures were performed for each set of batches in order for the data to be used for statistical analysis. The fermentation conditions were optimized for each different biocatalyst in terms of (a) fermentation time (24 h, 48 h, 72 h, and 96 h), (b) pH (4, 5, 6, and 7), (c) temperature (25 °C, 30 °C, 35 °C, and 40 °C), and (d) inoculum concentration (2.5%, 5%, 7.5%, and 10%). All cultures were performed under microaerophilic conditions and with constant agitation of 150 rpm in a rotary shaker. At the end of each experiment the fermentation broth was centrifuged (6000 rpm, 10 min) and the concentrations of ethanol and remaining sugars were estimated in the supernatant.

#### 2.3.3. Production of Bioethanol in a 7-L Fermenter

Fermentation was carried out in a BioFlo/CelliGen 115 fermenter provided by New Brunswick Co., New Jersy, USA, with all necessary controls. The reactor had a 7-L capacity and the working volume was 3 L. The fermenter was equipped with an agitator, pH, and temperature control system. The fermenter was cleaned and steam-sterilized at 121 °C for 20 min. Three liters of hydrolyzed and diluted SBW (32 g/L of sugar) was transferred to the fermenter and inoculated with *S. cerevisiae* KX008611, *P. barkeri* KX008612, and *C. intermedia* KX008614 (singly or in a consortium) as mentioned in batch culture procedures in a concentration of 5%. The temperature of fermentation was maintained at 30 ± 1 °C and the pH was regulated at 5. The agitator speed remained constant throughout the experiment at 150 rpm. The reactor was maintained under microaerophilic conditions. Samples were collected after 72 h of fermentation for the estimation of sugars and ethanol concentrations.

### 2.4. Analytical Techniques

The concentration of reducing sugars was estimated by the dinitrosalicylic acid method [24]. Starch was quantified via a total starch assay kit (Megazyme). Ethanol was

quantified via gas chromatography/mass spectrometry (GC/MS) (Agilent 6890 N/5975 B Germany) using a BR-SWax separation column (FS 30 m, 0.25 mm ID, 0.25 μm df). Helium at 2 mL/min was used as the carrier gas. The flow rates of high purity hydrogen and zero air were set at 30 and 300 mL/min, respectively. The temperature of the injection was set at 225, and 285 °C, respectively. The oven temperature was set initially at 45 °C for 2 min and then increased to the final temperature of 240 °C at the rate of 45 °C/min. The injection volume was 1 μL.

### 2.5. Statistical Analysis

All statistical analyses were performed using the SPSS 22.0 software (SPSS, 2013). The data were initially examined for their normality of distribution and homogeneity of variance. The significance of variations in the data for the effect of incubation time, temperature, pH, and inoculum concentration on ethanol production by yeast either singly or in a consortium in vitro was assessed using a two-way analysis of variance (ANOVA-2). The correlation between each parameter and ethanol productivity was calculated. One-way analysis of variance (ANOVA) was used to identify statistically significant differences among the means of consumed and unconsumed sugar and ethanol production in scale-up experiments. The least significant difference (LSD) test was used at $P < 0.05$ to identify the significant differences between the means among the treatments.

### 3. Results

The results showed that using co-cultures of yeast in either two or three different species increased the productivity of ethanol from the pretreated starchy-waste. The optimum incubation time for ethanol production was 72 h except for *Pichia barkeri* and *Candida intermedia,* which achieved their maximum production after 96 h. When the three yeasts were used in a consortium, the highest concentration of ethanol was attained after 72 h at 150.33 g/kg SBW (Table 1). Analysis of variance using the two-way ANOVA to study the effect of incubation time and application of yeast singly or in a consortium on ethanol production showed that means are significantly different at $P < 0.001$. So, at this level of probability, the null hypothesis was rejected, and the alternative hypothesis was accepted. There was a strong positive correlation between the incubation time and ethanol production from all treatments ($r = 0.855$) (Figure 1). The titer of produced ethanol was dependent on the incubation time by 73.1% ($R^2 = 0.731$).

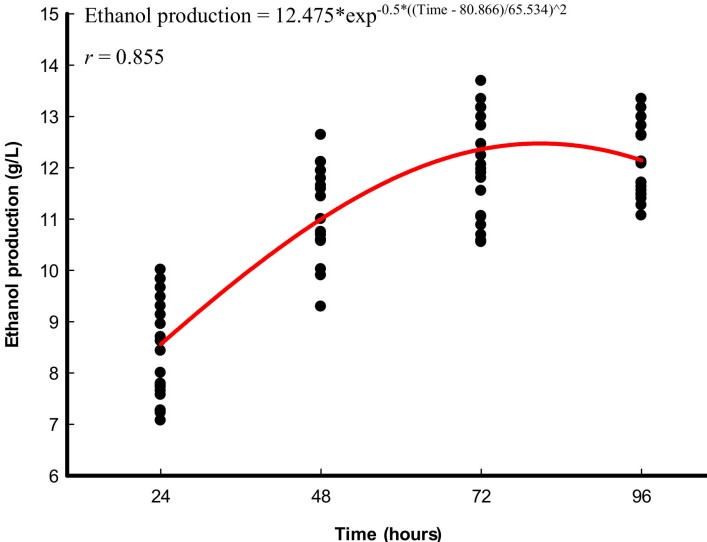

**Figure 1.** Correlation between incubation time (h) and ethanol production by yeasts (singly and in a consortium) from pretreated SBW in a batch culture at 30 °C, pH 5, and 150 rpm. Black dots represent the experimentally measured ethanol. Red line is the correlation tendle among values.

**Table 1.** Effect of fermentation time on ethanol yield, $Y_{E/SBW}$, from pretreated starchy biowaste (SBW) during fermentation of hydrolyzed SBW by yeast consortia in batch cultures at 30 °C, pH 5, and 150 rpm.

| Yeast | $Y_{E/SBW}$ (g/kg SBW) | | | | Mean (n = 3) ± SE |
| --- | --- | --- | --- | --- | --- |
| | Fermentation Time | | | | |
| | 24 h | 48 h | 72 h | 96 h | |
| *Saccharomyces cerevisiae* | 85.79 ± 0.49 | 120.64 ± 1.44 | 131.82 ± 1.17 | 131.00 ± 2.27 | 117.31 ± 5.68 b |
| *Pichia barkeri* | 80.63 ± 0.67 | 116.27 ± 2.60 | 122.21 ± 1.76 | 136.01 ± 3.20 | 113.78 ± 6.24 a |
| *Candida intermedia* | 85.27 ± 1.91 | 112.41 ± 4.73 | 120.12 ± 1.07 | 127.26 ± 1.62 | 111.26 ± 4.93 a |
| *S. cerevisiae* + *P. barkeri* | 94.69 ± 2.48 | 130.44 ± 1.14 | 135.64 ± 0.89 | 129.73 ± 1.64 | 122.62 ± 4.96 c |
| *S. cerevisiae* + *C. intermedia* | 99.14 ± 2.35 | 133.28 ± 1.72 | 144.42 ± 2.39 | 143.79 ± 1.13 | 130.16 ± 5.62 d |
| *P. barkeri* + *C. intermedia* | 107.67 ± 1.75 | 135.22 ± 0.64 | 146.44 ± 1.31 | 143.79 ± 1.13 | 133.28 ± 4.66 d |
| *S. cerevisiae* + *P. barkeri* + *C. intermedia* | 109.65 ± 1.74 | 137.84 ± 1.98 | 150.33 ± 1.72 | 149.02 ± 0.64 | 136.71 ± 498 e |
| Mean (n = 3) ±SE | 94.69 ± 2.42 a | 126.59 ± 2.21 b | 135.86 ± 2.51 c | 137.23 ± 1.83 c | 123.59 ± 2.19 |

Means followed by the same letter are not significantly different at *P* < 0.05 using Tukey's multiple test.

The temperature appeared as a main factor affecting the production of ethanol by the three yeasts either singly or in a consortium. The most appropriate temperature was 30–35 °C, at which the maximum concentration of ethanol was achieved. The highest productivity by both *Saccharomyces cerevisiae* and *Candida intermedia* was achieved at 35 °C, however, *P. barkeri* produced the highest concentration at 30 °C (Table 2). When the three yeasts were combined, the maximum productivity of ethanol was detected at 35 °C as 149.55 g/kg SBW. At both temperatures 25 °C and 40 °C, a considerable production of ethanol was detected but lower than the above-mentioned two temperatures (30 and 35 °C). Statistical analysis showed that there was a significant difference among the means of either yeast treatments or temperatures at *P* < 0.001 to indicate the real effect of the application of yeasts either singly or in a consortium and the incubation temperature on ethanol production. A very strong correlation between the ethanol productivity and the temperature was noted (r = 0.909) (Figure 2) indicating the dependence of ethanol productivity of the applied yeasts on temperature by 82.0% ($R^2$ = 0.82).

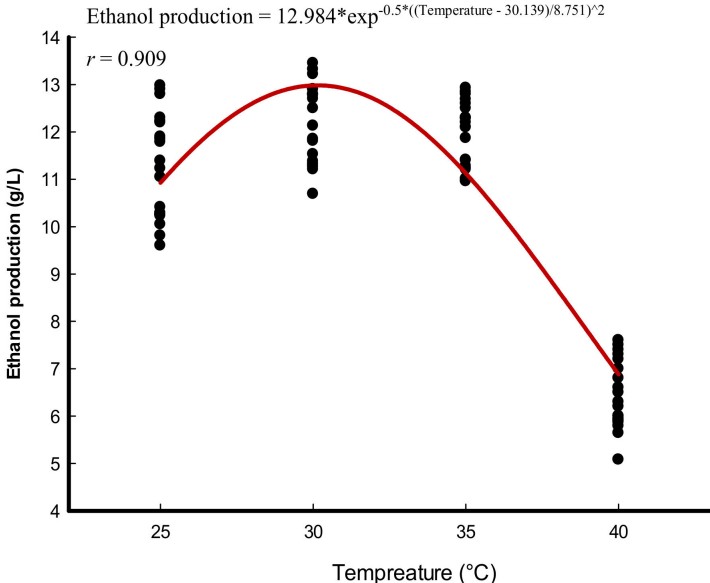

Ethanol production = $12.984 * \exp^{-0.5*((\text{Temperature} - 30.139)/8.751)^2}$

r = 0.909

**Figure 2.** Correlation between incubation temperature (°C) and ethanol production by yeasts (singly and in a consortium) from pretreated SBW in a batch culture at pH 5 and 150 rpm after 72 h. Black dots represent the experimentally measured ethanol. Red line is the correlation tendle among values.

**Table 2.** Effect of incubation temperature on ethanol yield, $Y_{E/SBW}$, from pretreated SBW during fermentation of hydrolyzed SBW by yeast consortia in a batch culture at pH 5 and 150 rpm after 72 h.

| Yeast | $Y_{E/SBW}$ (g/kg SBW) | | | | Mean (n = 3) ±SE |
|---|---|---|---|---|---|
| | Temperature (°C) | | | | |
| | 25 | 30 | 35 | 40 | |
| *Saccharomyces cerevisiae* | 113.39 ± 1.67 | 125.47 ± 2.84 | 126.89 ± 0.58 | 65.52 ± 1.12 | 107.82 ± 7.57 a |
| *Pichia barkeri* | 112.41 ± 2.36 | 130.59 ± 2.14 | 125.73 ± 1.26 | 65.63 ± 0.34 | 108.59 ± 7.78 a |
| *Candida intermedia* | 115.14 ± 0.15 | 126.73 ± 0.55 | 130.63 ± 3.91 | 63.53 ± 3.28 | 109.01 ± 8.78 a |
| *S. cerevisiae* + *P. barkeri* | 125.91 ± 1.10 | 140.95 ± 2.43 | 139.79 ± 1.91 | 74.42 ± 0.99 | 120.27 ± 8.21 b |
| *S. cerevisiae* + *C. intermedia* | 132.91 ± 0.36 | 142.86 ± 1.35 | 136.87 ± 0.65 | 71.80 ± 3.43 | 121.11 ± 8.69 b |
| *P. barkeri* + *C. intermedia* | 137.36 ± 0.33 | 143.60 ± 1.65 | 139.86 ± 0.99 | 78.91 ± 4.16 | 124.93 ± 8.09 c |
| *S. cerevisiae* + *P. barkeri* + *C. intermedia* | 144.65 ± 0.58 | 149.55 ± 0.75 | 143.86 ± 0.79 | 83.02 ± 1.29 | 130.27 ± 8.26 d |
| Mean (n = 3) ±SE | 125.97 ± 2.63 b | 137.11 ± 2.02 c | 134.80 ± 1.57 c | 71.83 ± 1.73 a | 117.43 ± 3.09 |

Means followed by the same letter are not significantly different at $P < 0.05$ using Tukey's multiple test.

The experimental results clearly demonstrated that pH 5 was the most appropriate pH for ethanol production by all yeasts from SBW. At this pH, the highest productivity was noticed when the three yeasts were used in a consortium (147.72 g/kg SBW). Using two-way ANOVA analysis, the alternative hypothesis was supported, and the null hypothesis was rejected at $P < 0.001$. There was a significant difference among means obtained from pH treatments (Table 3). Regarding the means obtained from yeast application, there was a significant difference among the co-cultures of two strains with those of three strains of yeast compared with the monocultures. There was a high positive correlation between ethanol concentration and the pH of the fermentation medium ($r = 0.84$) (Figure 3) which indicates the dependence of ethanol concentration on pH by 70% ($R^2 = 0.701$).

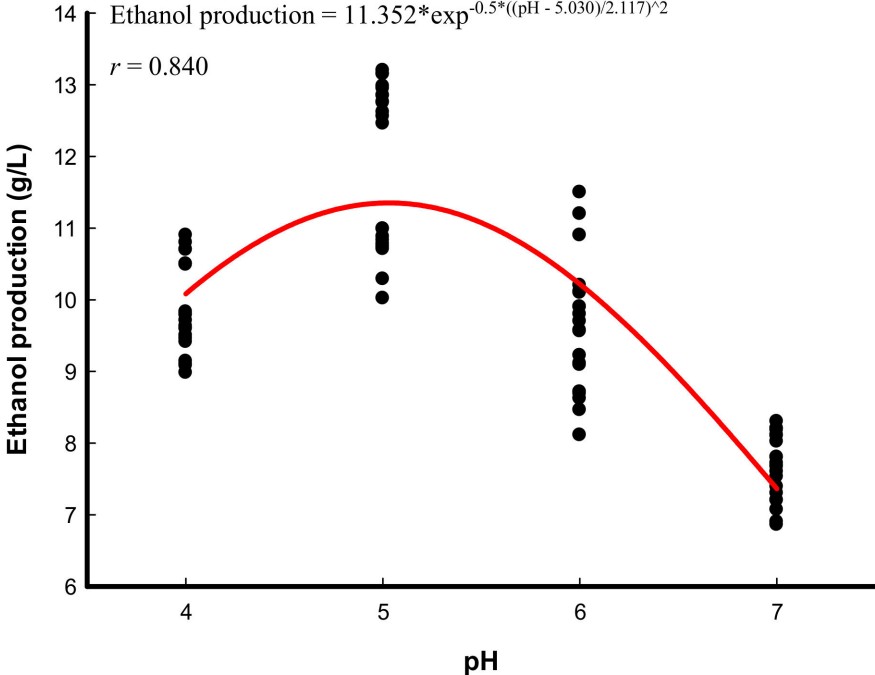

**Figure 3.** Correlation between initial pH and ethanol production by yeasts (singly and in a consortium) from pretreated SBW in a batch culture at 30 °C and 150 rpm after 72 h. Black dots represent the experimentally measured ethanol. Red line is the correlation tendle among values.

**Table 3.** Effect of initial pH on ethanol yield, $Y_{E/SBW}$, from pretreated SBW during fermentation of hydrolyzed SBW by yeast consortia in a batch culture at 30 °C and 150 rpm after 72 h.

| Yeast | $Y_{E/SBW}$ (g/kg SBW) | | | | Mean (n = 3) ±SE |
|---|---|---|---|---|---|
| | pH | | | | |
| | 4 | 5 | 6 | 7 | |
| *Saccharomyces cerevisiae* | 107.96 ± 1.36 | 117.69 ± 2.64 | 96.71 ± 0.90 | 83.36 ± 2.08 | 101.43 ± 3.94 a |
| *Pichia barkeri* | 101.57 ± 0.41 | 119.89 ± 2.33 | 107.67 ± 3.24 | 86.50 ± 2.04 | 103.91 ± 3.75 ab |
| *Candida intermedia* | 110.02 ± 4.37 | 122.29 ± 0.53 | 105.50 ± 2.82 | 91.59 ± 0.31 | 107.35 ± 3.49 cd |
| *S. cerevisiae + P. barkeri* | 107.22 ± 0.87 | 141.51 ± 0.97 | 95.14 ± 2.10 | 80.14 ± 2.96 | 106.00 ± 6.87 bc |
| *S. cerevisiae + C. intermedia* | 108.45 ± 0.75 | 143.45 ± 1.31 | 108.67 ± 0.78 | 81.15 ± 0.37 | 110.43 ± 6.68 d |
| *P. barkeri + C. intermedia* | 118.55 ± 0.75 | 144.27 ± 0.75 | 112.94 ± 0.99 | 86.76 ± 0.75 | 115.63 ± 6.17 e |
| *S. cerevisiae + P. barkeri + C. intermedia* | 121.17 ± 0.65 | 147.72 ± 0.19 | 125.65 ± 1.94 | 92.00 ± 0.65 | 121.63 ± 6.00 f |
| Mean (n = 3) ±SE | 110.70 ± 1.53 c | 133.83 ± 2.77 d | 107.47 ± 2.22 b | 85.93 ± 1.10 a | 109.48 ± 2.11 |

Means followed by the same letter are not significantly different at $P < 0.05$ using Tukey's multiple test.

The inoculum concentration had a clear effect on the ethanol production of the three yeasts. It was shown that when the yeasts were applied singly, there was not a significant difference between the concentration of 5% or 7.5%. However, when the yeasts were used in a mix, the concentration of 5% was the best one that induced the maximum production of ethanol (Table 4). At $P < 0.001$, the null hypothesis was rejected, and the alternative hypothesis was accepted to indicate the significant difference among the means of treatments. There was a significant difference among the means of yeasts when they were applied in a mix, even in dual culture. There was a positive correlation between ethanol production by the applied yeasts and the inoculum concentration ($r = 0.785$) (Figure 4). The ethanol productivity depends on the inoculum concentration by 0.62% ($R^2 = 0.616$).

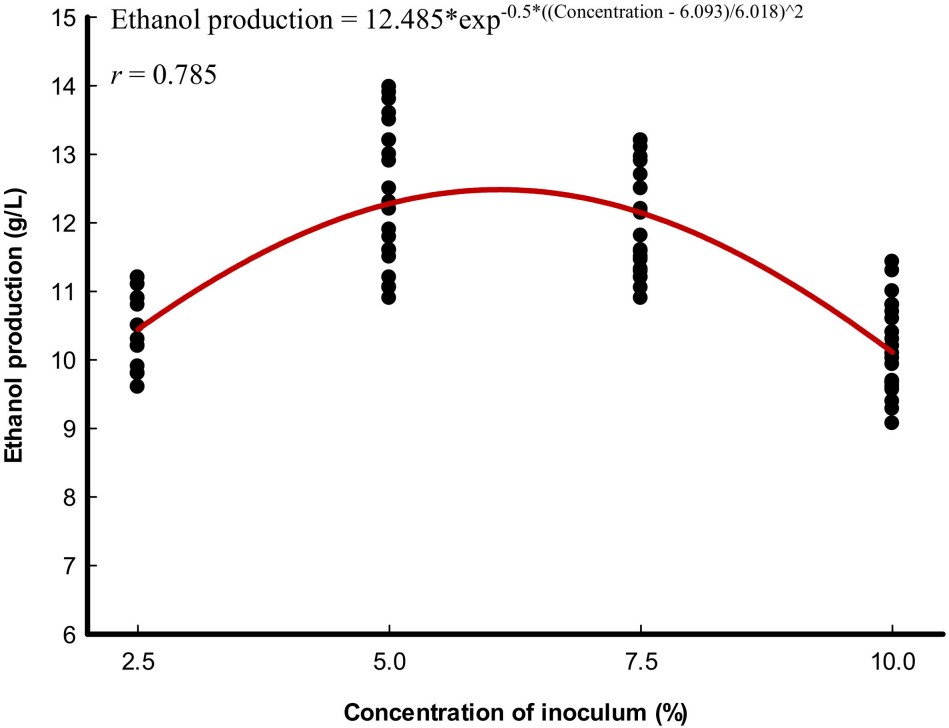

**Figure 4.** Correlation between inoculum concentration and ethanol production by yeasts (singly and in a consortium) from pretreated SBW in a batch culture at 30 °C, pH 5, and 150 rpm after 72 h. Black dots represent the experimentally measured ethanol. Red line is the correlation tendle among values.

**Table 4.** Effect of inoculum on ethanol yield, $Y_{E/SBW}$, from pretreated SBW during fermentation of hydrolyzed SBW by yeast consortia in a batch culture at 30 °C, pH 5, and 150 rpm after 72 h.

| Yeast | $Y_{E/SBW}$ (g/kg SBW) | | | | |
|---|---|---|---|---|---|
| | Concentration of Inoculum (%) | | | | Mean (n = 3) ±SE |
| | 2.5 | 5 | 7.5 | 10 | |
| *Saccharomyces cerevisiae* | 111.44 ± 1.49 | 130.89 ± 1.35 | 130.33 ± 1.15 | 107.25 ± 1.00 | 119.98 ± 3.28 ab |
| *Pichia barkeri* | 109.57 ± 0.99 | 128.83 ± 2.45 | 128.94 ± 3.71 | 106.32 ± 2.79 | 118.42 ± 3.37 b |
| *Candida intermedia* | 110.69 ± 1.98 | 123.97 ± 0.97 | 124.53 ± 1.12 | 108.56 ± 2.63 | 116.94 ± 2.35 a |
| *S. cerevisiae + P. barkeri* | 115.93 ± 0.99 | 138.37 ± 0.99 | 128.72 ± 0.92 | 109.38 ± 1.56 | 123.10 ± 3.42 c |
| *S. cerevisiae + C. intermedia* | 121.92 ± 0.37 | 146.22 ± 0.99 | 139.12 ± 1.12 | 115.56 ± 0.65 | 130.70 ± 3.77 d |
| *P. barkeri + C. intermedia* | 121.54 ± 1.98 | 150.71 ± 1.35 | 145.10 ± 1.63 | 120.04 ± 0.65 | 134.35 ± 4.18 e |
| *S. cerevisiae + P. barkeri + C. intermedia* | 125.28 ± 0.37 | 155.87 ± 0.58 | 146.82 ± 0.78 | 123.78 ± 1.63 | 137.94 ± 4.18 f |
| Mean (n = 3) ±SE | 116.62 ± 1.37 b | 139.27 ± 2.53 c | 134.79 ± 1.91 b | 112.99 ± 1.53 a | 125.92 ± 1.55 |

Means followed by the same letter are not significantly different at $P < 0.05$ using Tukey's multiple test.

Data collected from the 7-L fermenter clearly demonstrated the efficiency of the yeast consortium on the sugar consumption (Figure 5a). The maximum consumption of the available sugar was 92.2% when the three yeasts were applied as a mixture inoculum (5%) at 30 °C, pH 5, and 150 rpm after 72 h of incubation. When the yeasts were coupled, the consumption of sugar was 88.7% to 89.3%, however, when they were applied singly the consumption of sugar decreased to 86.5–88.2%. Consequently, the remaining amount of unconsumed sugars was also affected by the number of strains that were used in each culture. (Figure 5b). There was a significant difference between the mean of treatments at $p < 0.05$.

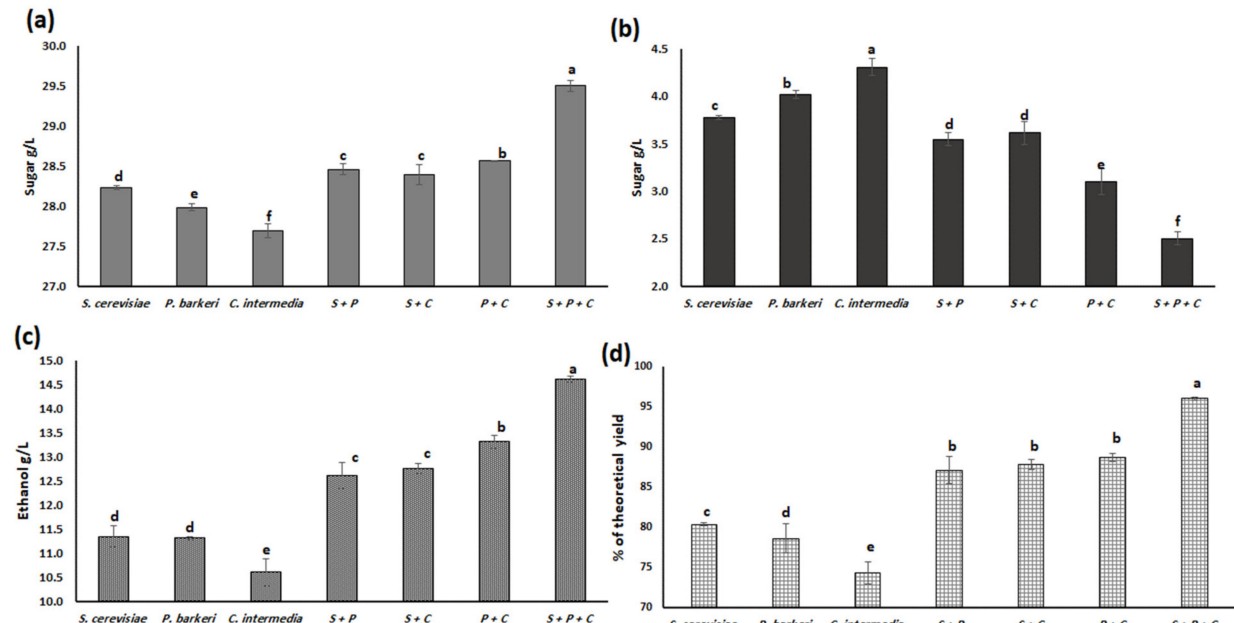

**Figure 5.** Consumed sugars (**a**), unconsumed sugars (**b**), ethanol titter (**c**), and fermentation efficiency of ethanol production (expressed as % from the theoretical expected yield) (**d**), by yeast singly and in consortium during fermentation in a scale-up experiment at 30 °C, pH 5, and 150 rpm after 72 h. Vertical bars represent the standard error (n = 3) and the different letters mean that the means are significantly different at $P < 0.05$ using one-way ANOVA analysis.

The production of ethanol in the scale-up experiment was greatly enhanced when the three yeasts were applied together (Figure 5c). The maximum ethanol concentration was 167.8 g/kg SBW in this case. The application of the yeasts in pairs or singly resulted

in significantly lower ethanol production than the consortium ($P < 0.05$). The lowest concentration of ethanol was obtained when *C. intermedia* was used as the sole biocatalyst, leading to the production of 121.8 g/kg SBW. *S. cerevisiae* and *P. barkeri* resulted in the production of similar titers of ethanol when they were used in monocultures, however, when they were used in co-cultures the produced ethanol titers varied. When we compared the real production of ethanol with the theoretically expected production, there was a good indication about the positive effect of the yeast consortium on the yeast efficiency. The yeast mixture gave 96.0% of the theoretically expected production (Figure 5d). This indicator was 87.0–88.6% of the theoretically expected production when the yeasts were used as couples, however, it lowered to 74.3–80.3% when the yeasts were used singly.

## 4. Discussion

Lately, great attention has been paid to the exploitation of FW that are discharged from households, restaurants, and food processing factories as fermentation substrates due to their abundancy, efficient bioconversion [5], but also because such an exploitation constitutes an eco-friendly process that may contribute greatly to greenhouse gas emissions savings and the concept of a circular economy [25]. Waste-rice comprises a considerable part of resultant waste throughout the globe and as such is worth investigating its potential use as fermentative feedstock [26–28]. On the other hand, bioethanol, as a main constituent of the current and future bioenergy, is a very important end product that can be generated from the fermentation of various biowastes [2,5,7–16]. In this context, the results of the current study clearly demonstrate the feasibility of the transformation of the pretreated waste-rice into ethanol using either monocultures or co-cultures of fermentative yeasts.

As indicated by the initial flask experiments, co-culturing was the optimal strategy for enhancing ethanol yields. Indeed, 150.33 g ethanol/kg of the SBW was produced when applying the co-culture of *S. cerevisiae*, *P. barkeri*, and *C. intermedia* after 72 h of incubation at 30 °C, pH 5, and 5% of the inoculum concentration. The co-culturing of different yeast strains from various substrates has indeed been proposed as advantageous for the efficiency of the ethanol production process compared to its pure cultures, leading to higher ethanol titers, maximum uptake of carbon sources, and better ethanol yields [14,17,18,29–32]. When the same fermentation process was carried out in a 7-L fermenter the productivity increased further, reaching 167.80 ± 0.49 g/kg SBW, which corresponds to an approximately 5% increase of yield compared to the yield obtained from the respective flask experiments. As previously reported by other studies [8,33,34] the current results confirmed that the optimization of fermentation conditions, such as fermentation time, temperature, pH, and inoculum size, is quite important for the maximization of ethanol titers, yields, and productivities, i.e., to fermentation outcomes that are crucial for the economic viability and overall sustainability of the process.

More specifically, focusing on the economic point of view, the fermentation time seems quite an important factor since it affects the productivity of the process [13]. Obtaining the maximum production of ethanol after 72 h by the applied yeast consortia makes the process more economically favorable compared to other research findings in which longer fermentation times are reported for the achievement of maximum bioconversion and ethanol production [35,36].

Regarding the pH of the fermentation, the application of a slightly acidic medium (pH 5) was proven to be the most appropriate pH for ethanol production by the selected yeasts either singly or in a consortium. It should be mentioned that all three yeast strains used in the current study are weak acidophilic and can tolerate pH values ranging from 4 to 7. Nevertheless, pH 5 induced the maximum ethanol production, a finding that is in agreement with previous studies. Narendranath and Power [37] found that the optimum pH for yeast growth and ethanol production by *S. cerevisiae* was pH 4.9, whereas Pramanik [38] reported that the maximum ethanol production by *S. cerevisiae* was achieved for pH 4.25–5.0. In a study by Pramanik [38], it was argued that the activity of the yeast species at pH 3.75 was decreased and this was attributed to the inability of the activation

of the fermentation enzymes at such a low pH. On the other hand, at the higher pH values tested, the low ethanol production and sugar conversion values could be due to the formation of undesired products like glycerol and organic acids [36]. According to other studies, many yeasts perform optimal fermentations at acidic pH, with values 5.0–5.2 being optimal for efficient fermentation, but brewing and distilling strains are capable of good growth at the pH range of approximately 3.5 to 6.0 [39]. As reported in that study, $H^+$ ions are excreted by yeast during any fermentation, resulting in a pH decline in the media. Nevertheless, the maintenance of the optimal pH in the medium is of the utmost importance when higher ethanol production is the target of the fermentation [18].

As it concerns the effect of the temperature on the efficiency of the process, the most appropriate temperature, at which all three yeasts produced the highest ethanol titter was 30–35 °C. This indicates the mesophilic nature of the tested yeasts. Determination of the appropriate temperature of fermentation is the crucial factor that affects the productivity of ethanol [40]. The temperature has a direct effect on the biochemical reactions and metabolism of yeasts because the enzyme activities are suppressed at low temperature and the higher temperature affects the alcohol production because of cell death [41]. Moreover, efficient ethanol production at mesophilic conditions is favorable for the economic viability of the process since it requires less energy for the maintenance of optimal fermentation, this is especially useful in countries with temperate climatic conditions.

With respect to the inoculum size, it was observed that inoculation with 5% induced the maximum ethanol production by all yeasts. However, in some cases, the inoculation with 7.5% seemed to further increase the productivity of ethanol, though without having a statistically significant difference with the inoculum of 5%. So, from an economic point of view, it seems preferable to use an inoculum size of no more than 5%, since such an approach requires a smaller scale of pre-culturing. It can be assumed that the increase in the inoculum concentration induces the rate of consumption of the sugars until a definite concentration of 5%. However, further increase of the inoculum size did not seem to enhance the fermentation process since it led to the exhaustion of the substrate, as has also been reported by previous studies [42,43].

The results obtained from the scale-up experiment were consistent with those obtained from the flask experiments that support the validity of the assumption that mixing more than one biocatalyst in one inoculum could enhance the ethanol productivity from the biowaste. The results of the current study demonstrated that when the three yeasts were applied in a consortium, the ethanol yield increased to $167.80 \pm 0.49$ g/kg SBW, which is statistically higher than the yield that was achieved during the respective flask experiment. The increase in the ethanol concentration in a 7-L fermenter could be related to various parameters that are affected by the scaling-up of a fermentation process such as the improved agitation and pH control. Indeed, in the 7-L-fermenter agitation was performed via mechanical stirring and not in a rotary shaker as in the flask experiments, resulting in more efficient mixing of the fermentation broth and consequently to better availability of the nutrients to the biocatalysts, leading thus to better productivities and yields [44]. Moreover, better mixing during the fermentation is reported to allow for more effective gas-liquid mass transfer [45] leading thus to better exchange of oxygen and fermentation gases [46] and also for more efficient heat transfer [47], which is expected to ensure optimal distribution of heat throughout the volume of the reactor. All the above are expected to ensure the maintenance of more homogeneous conditions in the fermenter that favor the fermentation performance of the yeasts, resulting in the increase of the ethanol yield. In addition to the most efficient agitation, the controlled operation of the fermenter allows the continuous adjustment of the pH to the optimum value of 5, preventing its possible drop or variations during the fermentation process. Continuous maintenance of pH at the optimum value is expected to have a positive effect on ethanol yield and therefore can correlate to the increased yields that were achieved in the fermenter experiments, compared to the flask experiments.

## 5. Conclusions

In previous decades, there have been numerous studies that have investigated the optimization of fermentation conditions for the production of bioethanol from many different waste streams and microbial strains. In this context, in the current study, the successful application of a biocatalytic yeast consortium was confirmed to ferment efficiently the SBW leading to increased ethanol production compared to the single-use of each yeast strain. The effect of the main fermentation parameters was investigated and the optimal conditions were determined for small-scale flask experiments. The subsequent scale-up of the process, applying the optimal conditions for all different yeast consortia and monocultures verified the observation of the superiority of the co-culturing of all three strains, leading also to even better performance and achieving a higher ethanol yield. As such, this finding confirms that for the design of truly sustainable processes, an investigation should not be limited to the identification of promising biocatalysts and the determination of optimal conditions but should be also expanded to the thorough study and implementation of the gradual increase of the scale of the process.

**Author Contributions:** M.H. and S.A.A.; conceptualization, Y.S.M. and T.A.Y.A.; methodology, M.H., S.A.A. and I.N.; data curation, M.H. and Y.S.M.; writing—original draft preparation, I.N. and G.L.; writing—review and editing, M.H. and G.L.; project administration. All authors have read and agreed to the published version of the manuscript.

**Funding:** This research was funded by the Deputyship for Research & Innovation, Ministry of Education in Saudi Arabia, under project No 327: "CiClic—Enhanced pilot scale bioethanol production from food wastes using novel biocatalysts".

**Acknowledgments:** The authors extend their appreciation to the Deputyship for Research & Innovation, Ministry of Education in Saudi Arabia for funding this research work through project number 327.

**Conflicts of Interest:** The authors declare no conflict of interest.

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
