# Peer review of "On the Optimization of Fermentation Conditions for Enhanced Bioethanol Yields from Starchy Biowaste via Yeast Co-Cultures"

_sustainability, doi:10.3390/su13041890_

Round 1

Reviewer 1 Report

Hashem and Co-workers submitted the paper entitled “On the optimization of fermentation conditions for enhanced bioethanol yields from starchy biowaste via yeast co-cultures” to publish in “Sustainability (I. F = 2.576)”. In this paper, they have optimize the fermentation conditions by using different yeast consortium and co-culture tactic. The work seems to be an impressive one and presentation is nice, thus can be published after addressing the queries.

1. Why the author chosen only “Saccharomyces cerevisiae, Pichia barkeri and Candida intermedia” for this study? How about Saccharomyces cerevisiae KL17, which has the great Ethanol productivity (3. 46 g/L/h)?. Clear explanation in the introduction part is needed for the selection of those yeast consortium for co-culture tactic.

2. How about ethanol productivity at 15⁰C and at incubation time 12 and 120 hours?

3. Reference section requires updation on similar studies.

Author Response

Hashem and Co-workers submitted the paper entitled “On the optimization of fermentation conditions for enhanced bioethanol yields from starchy biowaste via yeast co-cultures” to publish in “Sustainability (I. F = 2.576)”. In this paper, they have optimize the fermentation conditions by using different yeast consortium and co-culture tactic. The work seems to be an impressive one and presentation is nice, thus can be published after addressing the queries.

Thank you for your comments.

  1. Why the author chosen only “Saccharomyces cerevisiae, Pichia barkeri and Candida intermedia” for this study? How about Saccharomyces cerevisiae KL17, which has the great Ethanol productivity (3. 46 g/L/h)?. Clear explanation in the introduction part is needed for the selection of those yeast consortium for co-culture tactic.

Response: Thank you for your comment. We certainly do agree that there are several ethanologenic yeasts strains that are reported to produce extremely high ethanol titers from various simple and complex substrates. Among them S.cerevisiae KL17 is indeed reported to have great ethanol productivity. However, the specific strain was isolated from soil and is reported to produce ethanol from galactose. As such the use of such a strain was out of the scope of our study. As mentioned in the paragraph 2.3.1. the selected yeasts strains were isolated from naturally fermented waste rice i.e. a substrate similar to the waste that was used as feedstock in the present study. The strategy of selecting fermentation microorganisms from similar substrates to the targeted feedstock is considered to be advantageous for the efficiency of the process since the strains are expected to adapt easily to the new substrate.  Moreover, during the isolation process several different strains were isolated and their fermentation capacity of glucose is assessed. The selection of the strains that were used in the present study was made based on the higher ethanol production compared to other strains. The comparative application of the selected strains in mono-cultures and co-cultures was also based in the assumption that microorganisms in consortia tend to develop synergistic relationships in order to survive better in a competitive environment and therefore may lead to more efficient exploitation of the supplied substrates, and in the case of the present study case to more efficient ethanol production process as mentioned in lines 59-60 and 67-72 of the initial MS. The above are briefly explained in the Introduction section of the revised MS, according to your suggestion.

  1. How about ethanol productivity at 15 ⁰C and at incubation time 12 and 120 hours?

Response: Thank you for your comment. The selection of upper and lower limits for the tested parameters i.e. temperature, incubation time and pH was made based on previous studies of the implicated in the study research teams, on the literature and also considering the viability of the proposed  process and the origin of the yeasts which was isolated from fermented wasted-rice at mesophilic conditions. Based on the above in terms of the temperature in was not considered necessary to test temperatures below 20oC, or above 40oC (typical mesophilic range is 20-40 oC). In terms of the incubation time based on the mean μmax of similar yeasts and the initial concentration sugars used for all fermentation tests, it was estimated that for incubation times below 24h, no sufficient amount of substrate would be bio- converted. Finally, it was estimated that the inculcation time of 4 days (96h) is long enough based on the literature and for inculcation times above that the operation cost of the fermenter might be limiting for the viability of the process.

  1. Reference section requires updation on similar studies.

Response: Thank you for your comment. The reference list was updated according to your suggestion by adding six more references.

Reviewer 2 Report

The paper is devoted to the studies of impact of the type of yeast consortium used during bioethanol production from starchy biowastes, and to determination of the optimal fermentation conditions for bioethanol production. Three different yeast strains, were used in mono- and co-cultures for fermentation of pre-treated wasted rice. The dependence of ethanol yield upon  fermentation conditions was investigated in small scale batch cultures and subsequently for scaling up and validation of the process in larger fermenter. It was demonstrated that co-culturing of yeasts either in couples or triples significantly affects ethanol yield leading to an increase up to almost 170g/kg of biowaste. The results are carefully elaborated by means of statistical analysis. The aim of the work is well defined, and experiments together with their interpretation are performed professionally.

Results are meaningful.

The paper requires English edition.

Some of the unfortunate examples are shown below:

 32 bioenergy with lowering the cost effective through using low-cost substrates, efficient fermentative organisms, and optimization the conditions for obtaining the maximum yield

CAAGACGG-3´) [12] and were submitted to the GenBank and as new yeast strains

104 statistically significant 149 differences among the means of consumed and unconsumed sugar and ethanol production of ethanol in pilot experiments.

162 There was a strong positive coloration between incubation time and ethanol production from all  treatments (r = 0.855)

163  The connection of the produced ethanol is dependent on the incubation  time by 73.1% (R2 = 0.731).

183  correlation between the ethanol productivity and the temperature and ethanol productivity (r = 0.909)

200         double and tribble mixing of yeast

233    Consequently, the unconsumed sugar was affected by the application of the yeasts either singly or in combination                the meaning of this sentence is not clear

246   they applied singly, however, when they applied a paired the quantities of the product was varied.

Author Response

The paper is devoted to the studies of impact of the type of yeast consortium used during bioethanol production from starchy biowastes, and to determination of the optimal fermentation conditions for bioethanol production. Three different yeast strains, were used in mono- and co-cultures for fermentation of pre-treated wasted rice. The dependence of ethanol yield upon  fermentation conditions was investigated in small scale batch cultures and subsequently for scaling up and validation of the process in larger fermenter. It was demonstrated that co-culturing of yeasts either in couples or triples significantly affects ethanol yield leading to an increase up to almost 170g/kg of biowaste. The results are carefully elaborated by means of statistical analysis. The aim of the work is well defined, and experiments together with their interpretation are performed professionally.

Results are meaningful.

Response: Thank you for your comments.

The paper requires English edition.

Response: Thank you for your comment. English language was revised throughout the MS and the specific grammatical and syntactic errors were corrected according to your suggestions.

Some of the unfortunate examples are shown below:

 32 bioenergy with lowering the cost effective through using low-cost substrates, efficient fermentative organisms, and optimization the conditions for obtaining the maximum yield

Response: The sentence was revised.

CAAGACGG-3´) [12] and were submitted to the GenBank and as new yeast strains

Response: ‘and’ was deleted.

149 differences among the means of consumed and unconsumed sugar and ethanol production of ethanol in pilot experiments.

Response: The sentence was revised.

162 There was a strong positive coloration between incubation time and ethanol production from all  treatments (r = 0.855)

Response: The term was corrected.

163  The connection of the produced ethanol is dependent on the incubation  time by 73.1% (R2 = 0.731).

Response: The sentence was revised.

183  correlation between the ethanol productivity and the temperature and ethanol productivity (r = 0.909)

Response: The sentence was revised.

200         double and tribble mixing of yeast

Response: The sentence was revised.

233    Consequently, the unconsumed sugar was affected by the application of the yeasts either singly or in combination                the meaning of this sentence is not clear

 Response: The sentence was revised.

246   they applied singly, however, when they applied a paired the quantities of the product was varied.

Response: The sentence was revised.

Reviewer 3 Report

The present work is up to date and interesting since it deals with the fermentative process with three yeast strains and their combination for the production of bioethanol from starchy food waste and the approach might be applied to many different waste streams.

Overall, the work is clearly reported and the results are properly supported.

Indeed, I just have one relevant comment to be addressed to improve the manuscript. The authors should not refer to the fermentation in the 7-L fermenter as a pilot experiment (Line 241). In addition, this might be considered as a scale up experiment relatively to the previous flask experiments but there is also a change in the bioreactor configuration applied. For instance, the process modifications by introducing mechanical stirring (instead of orbital shaking), and also pH control, might be important and should be considered in the discussion with more relevancy than the effect of increased scale.

Also, the authors should carefully review all the manuscript to avoid some small spelling errors, such as the use of “tribble” instead of “triple” in Line 199.

Providing the authors address these minor comments, to my opinion the manuscript might be acceptable for publication.

Author Response

REVIEWER #3

The present work is up to date and interesting since it deals with the fermentative process with three yeast strains and their combination for the production of bioethanol from starchy food waste and the approach might be applied to many different waste streams.

Overall, the work is clearly reported and the results are properly supported.

Response: Thank you for your kind comments.

Indeed, I just have one relevant comment to be addressed to improve the manuscript. The authors should not refer to the fermentation in the 7-L fermenter as a pilot experiment (Line 241). In addition, this might be considered as a scale up experiment relatively to the previous flask experiments but there is also a change in the bioreactor configuration applied. For instance, the process modifications by introducing mechanical stirring (instead of orbital shaking), and also pH control, might be important and should be considered in the discussion with more relevancy than the effect of increased scale.

Response: Thank you for your comments. According to your suggestion the word “pilot” has been removed from the MS, and “scale-up” is used in all cases when referring to the experiments in the 7-L fermenter (Lines 156-157 and 247 of the revised MS). Moreover, the discussion about the enhanced performance of the scale-up experiment was rewritten taking into account your suggestions about the agitation type and pH control (Lines 327-344 of the revised MS)

Also, the authors should carefully review all the manuscript to avoid some small spelling errors, such as the use of “tribble” instead of “triple” in Line 199.

Response: Thank you for your comments. The English language was revised throughout the MS and grammatical and spelling errors were corrected.

Providing the authors address these minor comments, to my opinion the manuscript might be acceptable for publication.

Response: We would like to thank you for your positive evaluation.
